# Effect of Phase Fluctuation on the Proper Operation of Smart Gear Health Monitoring System

**DOI:** 10.3390/s22093231

**Published:** 2022-04-22

**Authors:** Thanh-Tung Mac, Daisuke Iba, Yusuke Matsushita, Seiya Mukai, Nanako Miura, Arata Masuda, Ichiro Moriwaki

**Affiliations:** Department of Mechanical Engineering, Kyoto Institute of Technology, Goshokaidou-cho, Matsugasaki, Sakyo-ku, Kyoto 606-8585, Japan; iba@kit.ac.jp (D.I.); matsushita@pml.mech.kit.ac.jp (Y.M.); mukai@pml.mech.kit.ac.jp (S.M.); miura-n@kit.ac.jp (N.M.); masuda@kit.ac.jp (A.M.); ichi@mech.kit.ac.jp (I.M.)

**Keywords:** smart gear, gear health monitoring, printed sensor, printed spiral antenna, plastic gear, magnetically coupled circuits

## Abstract

A smart gear sensor system has been developed for the condition monitoring of gear. This system includes a smart gear—the operation gear and a monitoring antenna. The analysis of the return loss of the monitoring antenna magnetically coupled with the smart gear gives the health condition of the gear. This research considers the effects of the distance and phase fluctuations between two components on the magnetic resonant return loss. The impacts of phase fluctuations include both static and high-speed conditions. Two experimental rigs have been built for the two cases. The coupling distance and static phase fluctuation are conducted via the first experimental rig. The second experimental rig performs both the coupling distance and phase fluctuation effect simultaneously while the smart gear rotates at high speed. During each test, the monitoring antenna return loss is captured thanks to a network analyzer. Analysis of the return loss data demonstrates that both the coupling distance and the change of the phase angle at the static condition and high speed have influenced the resonant return loss of the monitoring antenna. These findings are meaningful to the authors for evaluating and improving the accuracy of this gear health monitoring technique.

## 1. Introduction

Today, period maintenance is compulsory to remain sustainable in mechanical systems such as a car or a machine. However, in high-performance mechanical systems such as a helicopter or bullet trains, the maintenance strategies are inadequate to assure the reliability and safety of these high-performance systems [1]. Therefore, the condition-monitoring technique is considered an additional solution for the reliability and safety of these systems. As a result, the power transmission in which a gear is an indispensable component becomes a favorite target for condition-monitoring techniques. So far, according to the literature review, numerous solutions for gear health monitoring have been published. These solutions largely include the analysis of vibrational data, acoustic emission, a combination of vibrational analysis, and relevant component considerations such as lubricant condition, or the latest trending technology that is detection using sensors. Regarding the analysis of vibration response, typical techniques are introduced and reviewed by [2,3]. In fact, to extract the vibration data from a power transmission system, piezo accelerometers or sensors are mainly utilized. These sensing devices are often attached to the housing of the power transmission or gearbox. In this way, the characteristic vibration data of the gearbox can be collected completely. In [2,3], the authors indicated that techniques based on Fourier transform and Wavelet transform are the most popular techniques applied for vibrational signal processing. To improve the effectiveness of the pure vibration signal processing, an advanced technique that combines the vibrational signal processing with other methods are also introduced by [4,5,6,7]. Ref. [4] aimed to detect failures occurring in plastic gears during the gear operation by using the vibrational signal processing and a neural network technique. Refs. [5,6] demonstrated that acoustic emission is an effective tool for condition monitoring of gear teeth in a gear engagement set and bearing failures in a helicopter gearbox, respectively. To improve the quality of the detection of the progression of gear macro-pitting occurring in a gearbox, ref. [7] proposed a method that used the vibrational signal processing method in combination with the online visual monitoring for the particle in lubricants. Generally, even though the results of these studies demonstrated that the relevant proposed methods were helpful as the condition-monitoring tools of the gears, the vibration-based, condition-monitoring methods are known to be complicated methods. Since the vibration data contains the characteristic information of not only the targeted gear, but also from other components, noise removal is a challenge for the system operator. Moreover, the vibrational signal processing technique also requires the resources of the system and the time and effort of the operator. Overall, the described limitations can be considered common for any vibration-based, condition-monitoring method.

Furthermore, the solution for collecting the characteristic data of gears using sensitive sensors has been implemented by numerous researchers. Research by [8] showed the applications of wireless sensors for gear health monitoring purposes. A type of sensor called temperature nodes integrate into the housing of the gearbox. The data obtained wirelessly from the sensors are useful for multi-failures recognitions such as tooth fault or misalignment of the shaft. Although the results indicate that the wireless sensors are effective for the gear health monitoring technique, the power supply is the main concern of this integrated wireless sensor. Micro-electromechanical systems (MEMS) sensors were selected by [9] as a highly accurate solution for monitoring the failure of gear teeth in a gearbox of a helicopter. Specifically, the MEMS sensors are directly integrated on the surface of the gears. Therefore, the vibration response of the gear measured by these MEMS sensors has a higher quality for the gear failure detection analysis, since it mostly contains the characteristic information of the targeted gear with less noise. This is also a vibration-based, condition-monitoring technique.

Therefore, to address the limitations of vibration-based condition monitoring, the authors have developed a new gear health monitoring technique. Similar to [9], this technique also integrates sensors directly on the surface of the operation gear. Compared to the application of the MEMS sensors, our sensing technique works based on magnetic coupling, therefore the fabrication of sensors on the operation gear surface can be quick and simple. Specifically, there is no requirement for a battery or power supply or any sophisticated module to assist the feature of the sensor. The name of our gear health monitoring technique is so-called “a wireless smart gear system”. This system consists of an operation gear—the smart gear and a signal receiving terminal— and a monitoring antenna. In this system, a laser sintering technique is applied to create both the antenna pattern and sensor pattern on the monitoring antenna and the smart gear, respectively. The term “smart gear” is named for the operation gear in our gear health monitoring system that has a sensor layer on its surface. Thanks to this sensor layer, any crack or break that occurs on the physical operation gear can be monitored from an external terminal. In other words, the operation gear becomes “smarter” than the normal operation gear from other systems. Regarding the laser sintering technique, ref. [10] provided descriptions and devices that we have used to create the sintered sensor layer. A 3.5 [W] laser module has been integrated into a four-axis CNC machine Roland MDX-40A to sinter a silver nanoparticle ink NPS-J to a surface, where the monitoring antenna surface and the MC 901 plastic spur gear surface are located. In addition, since the sintered sensor layer is conductive, the sintered layers on the monitoring antenna and smart gear form close electric circuits. These electric circuits are called the monitoring antenna circuit and smart gear sensor track circuit, respectively. Our previous research [11] revealed that magnetic coupling occurs between this monitoring antenna circuit and the smart gear sensor track circuit when the sensor track circuit of the smart gear is placed nearby and aligned with the monitoring antenna circuit during the connection of this monitoring antenna to a network analyzer. Due to the magnetic coupling, the return loss of the monitoring antenna will change from its single return loss to resonant return loss. The single return loss can be measured directly when the monitoring antenna is connected to the network analyzer via its connector. The difference between the resonant return loss and the single return loss is the shape of the first valley of the return loss chart. Particularly, the first valley of the single return loss has a one-peak shape, while the first valley of the resonant return loss has a two-peak shape. Based on the two-peak shape of the resonant return loss, ref. [11] also indicated that the change of this two-peak shape, which can be observed via a computer screen, is related to the physical conditions of the sensor track circuit of the smart gear. There were two physical conditions of the sensor track circuit that have been considered: healthy state, or uncracked, and unhealthy state, or cracked. The term “healthy” comes from the term “structural health monitoring”, which defines a normal structure—without any malfunction or damage. Similarly, the term “unhealthy” defines an abnormal structure—with malfunction or damage. In the scope of our project, a healthy gear refers to a smart gear that has no defects on its sensor track circuit, whereas an unhealthy gear refers to a deteriorated smart gear that has a crack or damage on its sensor track circuit due to the gear operation. Correspondingly, the healthy state and unhealthy state of the sensor track circuit of the smart gear result in two different two-peak shapes of the resonant return loss. Inversely, by recognizing the difference between these two two-peak shapes, the healthy and unhealthy states of the sensor track circuit of the smart gear, or the physical condition of the smart gear, can be figured out. This consequently affirms that the resonant return loss and its two-peaks shapes are pivotal to the gear failure recognition of our developing wireless smart gear system.

However, in a practical gear operation, the smart gear will rotate for the power transmission while the antenna is fixed, and the phase angle of the smart gear with respect to the monitoring antenna will change continuously. In accordance with this condition, the consideration of the influence of this phase angle change on the magnetic coupling is essential. Based on this consideration and its result, the authors can make a comparison and evaluate the potential error that can lead to any wrong diagnosis of the gear failure detection process. In a previous study [12], although the effect of phase fluctuation has been mentioned and considered by the authors, the result of this study can only show the effect when the phase angles were regularly changed at static conditions. Moreover, the samples are only two POM plates with sintered patterns on their surfaces. Thus, in this research, the authors focus on considering the effect of phase angles on the magnetic coupling signal comprehensively. Instead of POM plate samples, a practical smart gear system is utilized for the analysis and evaluation. This practical system includes a monitoring antenna with a connector and a completed smart gear with a healthy sensor chain. Experimental works are performed to evaluate the effect of phase fluctuation for both static and high-speed rotational conditions. Furthermore, the effect of the coupling distance that is familiar in a magnetic coupling system is also implemented and expressed as a supplement for the evaluation of the phase fluctuation effect in this research.

The content of this manuscript is based on a four-section structure. Initially, the first section—Introduction—declares the motivation and the objectives of this research. Next, the second section—Materials and Methods—provides an explanation of the materials and methods used in this paper. Then, the third section—Results and Discussion—shows all of the data of the experimental works and the analysis and discussion of the results and findings. Finally, the fourth section—Conclusion—summarizes the findings of this research work.

## 2. Materials and Methods

### 2.1. Smart Gear System and Materials

Figure 1 denotes a practical smart gear health monitoring system (Figure 1a) and its two major components, namely the monitoring antenna (Figure 1b) and the smart gear (Figure 1c), respectively. As referred to in Figure 1a, during the gear operation, the smart gear plays a role as the power transmission gear while the monitoring antenna is fixed in front of the smart gear surface where the sintered sensor track circuit is located. This setting enables the magnetic coupling between the monitoring antenna and the smart gear to occur even if the smart gear is in operation or stopped. Via a radio frequency (RF) cable, the monitoring antenna can retain its connection to a network analyzer. In this way, the resonant return loss of the system can be updated simultaneously with the gear operation. The smart gear is held firmly to the head of a shaft by two pins and a nut. The distance between the monitoring antenna and the smart gear or the coupling distance is designed as 1 mm. Based on the authors’ empirical experiences, the distance of 1 mm allows the system to work smoothly and safely, especially when the smart gear is broken due to the driving test, in which the debris of the smart gear is not stuck in this 1 mm gap of the coupling distance. Figure 1b,c is a practical sample for the experiments of this research. As mentioned in the introduction section, the laser sintering technique has been developed to create both the antenna pattern on the monitoring antenna and the sensor pattern on the smart gear by sintering a silver nanoparticle conductive ink NPS-J (Harima Chemicals Group, Inc., Tokyo, Japan) on the surfaces of these components. In this research, the antenna pattern in Figure 1b and the sensor pattern in Figure 1c are also fabricated by this laser-sintering technique. The base material of the monitoring antenna in Figure 1b is polyacetal or POM plastic, while the base material of the smart gear in Figure 1c is MC 901 nylon plastic. The selection of POM plastic for the fabrication of the monitoring antenna is that this material has excellent dimensional stability, low thermal expansion, and acetone resistance. Since the procedure of fabricating the monitoring antenna requires curing on a hot plate and cleaning the residual conductive ink with acetone, POM plastic is a reasonable selection. Moreover, POM plastic gear is typically used in our laboratory, and the authors have also previously considered making a smart gear using a POM gear. To improve the quality of the laser-sintering process, one layer of polyimide tape was applied to the surface of the POM plate. Moreover, the MC 901 nylon gear was selected for manufacturing the smart gear instead of a POM gear due to its advantageous physical properties compared to the POM gear.

Figure 2 provides specific dimensions of the patterns of the monitoring antenna and smart gear in Figure 1b,c. As can be seen, the monitoring antenna pattern in Figure 2a consists of two components, namely an open spiral antenna coil and an annular band, here and after called a ground (GND) part. According to Archimedes’ spiral, the open spiral coil has been designed with an initial radius of 12.5 mm, a line width of 0.5 mm, and a pitch of 1 mm. The inner and outer diameters of the GND part are 31 mm and 42 mm, respectively. In the case of the smart gear pattern, there are two holes in its antenna pattern, which is due to the pinhole of the practical smart gear. In fact, the authors’ experimental work has shown that the effect of the two cut holes on the magnetic coupling is unnoticeable. Therefore, one is able to state that the function of the antenna pattern on both the antenna and smart gear is identical. Compared to the monitoring antenna pattern, the difference between the smart gear pattern is the additional sensor chain surrounding the antenna pattern. Since the smart gear is also the operation gear, the tooth root area is considered to be the place with the most potential for a crack to occur. Hence, the sensor chain is designed to cover all the tooth root areas, as illustrated in Figure 2b. Overall, the resistance sensor chain is a conductive track that connects the open spiral coil and the GND parts of the antenna pattern to form a close sensor track circuit. As a result, when a crack appears on the smart gear due to damage, the conductivity of the sensor chain is changed. This physical change makes the characteristic properties of the close sensor track circuit change. As discussed in the introduction section, the different physical states of the close sensor track circuit on the smart gear result in the differences in the resonant return loss of the monitoring antenna. Moreover, via these differences, the authors can indicate the physical conditions of the smart gear. This also defines how the smart gear system works.

### 2.2. Method

Figure 3 provides a simple procedure of the experiments. In the first stage, a test rig is configured to consider the relevance between the distance and the coupling quality of the monitoring antenna and the smart gear. Specifically, the smart gear is kept concentrically and in parallel with the monitoring antenna while it can be translated vertically along the center axis. Since the monitoring is fixed, this setting means that the coupling distance between the smart gear and the antenna can be changed easily. The range is from 0 to 6 mm. As a result, the accomplishment of this test is the detailed relationship between the return loss forms of the system and the distance. As introduced in Section 2.1, the coupling distance for the experiment is 1 mm, based on the authors’ empirical experience, therefore knowing the behavior of the resonant return loss due to the effect of the coupling distance can help the authors to clarify and evaluate the empirical selection of 1 mm of coupling distance more solidly. Moreover, thanks to the rotatable top disk of the 3D stage that has 360 divisions of degrees, the static phase fluctuation effect of the smart gear can be considered simultaneously. Based on the alignment of two antenna patterns on the monitoring antenna and on the smart gear, the initial relative phase angle is equal to 0 degrees when all the open spiral coil and the GND part of the two antenna patterns are in parallel. Then, the phase angle of the smart gear is switched every 5 degrees with respect to the monitoring antenna. The coupling distance in this configuration is also 1 mm. Moreover, the resonant return loss of the magnetic coupling system is measured after the switching of the phase angle via the monitoring antenna and the network analyzer.

In the second stage, to drive the smart gear at high speed, another test rig has been configured. In this test rig, the monitoring antenna and the smart gear are also aligned concentrically. In contrast to the static phase fluctuation test rig, the smart gear is secured to the spindle of the electrically driven motor in this driving test rig while the monitoring antenna is fixed to a lever that has a grooved rail at its bottom. The grooved sliding rail allows the lever to carry the monitoring antenna and can slide along the rail of the test rig in the direction of the horizontal center axis of both the monitoring antenna and the smart gear. This type of designed test rig enables the coupling distance between the monitoring antenna and smart gear to be adjusted simply. In this stage, the consideration of the coupling distance when the smart gear is operating at high speed aims to indicate how the phase fluctuation at high speed affects the magnetic coupling at different degrees of the magnetic coupling magnitude. The coupling distance is changed from 1 mm to 6 mm by each 1 mm. The smart gear is sped up by a high-speed motor with a range of rotational velocity from 0 rpm to 3500 rpm. According to each specific value of the coupling distance, the rotational speed of the smart gear is adjusted from its designated minimum speed to the maximum speed correspondingly. The resonant return loss value of the monitoring antenna is captured right after the speed is modified due to to the network analyzer. The analysis and comparison of the return loss values show the effect of the phase fluctuation at high speed in detail. Based on the results, the authors can evaluate the most effective operation speed for the developing smart gear system so that the accuracy of this gear health monitoring technique can be enhanced.

#### 2.2.1. Sintering Condition for MC 901 Smart Gear

In reference to [10], to assure the quality of the sintered layer, it is necessary to treat the surface of the printing objects, namely the POM plate and an MC 901 nylon gear. Currently, to treat the POM plate surface, the authors apply one layer of polyimide tape on its surface. Practically, this application of polyimide remarkably improved the surface quality of the POM plate. The effectiveness of the polyimide layer in increasing the quality of the printed layer is also indicated in the aforementioned previous research. Nevertheless, the raw surface of the MC 901 gear shows that it cannot be enhanced by using just the polyimide layer as in the case of POM plate. On one hand, due to the remarkable high roughness of the gear side surface, the bonding quality between the polyimide tape and the gear surface has low quality. As illustrated in Figure 4, the roughness value *R_z_* of the raw surface of the MC 901 nylon gear is *R_z_* ≈ 142 µm. This value is measured by an Olympus LEXT OLS5000, a digital three-dimensional measuring laser microscope. Moreover, applying the polyimide layer on the gear side surface and modifying it to match the gear profile also takes time and effort. On the other hand, the surface condition of the raw MC 901 gear has also empirically proven that it is not suitable for a direct laser sintering process, since the conductivity of the sintered sensor layer is much lower than a similar sensor layer sintered on a POM plate with one layer of the polyimide tape.

In order to resolve the discussed limitation, an idea to improve the surface quality of the MC 901 gear has been proposed. Figure 5 illustrates a specialized device for the MC 901 gear surface treatment work. Originally, this device is a DVD polisher with one main DC electric motor on its top and a rotatable circular plate at the bottom. During the operation, a fabric circular plate is attached to the spindle of the DC electric motor while the targeted polishing object is placed on the lower rotatable plate. Since the center axis of the upper DC electric motor and the center axis of the lower rotatable plate are deflected to each other, the relative movement of the upper DC electric motor and the lower rotatable plate carrying a DVD can polish the surface of the DVD well. Aiming to use this operation feature for the MC 901 nylon gear surface treatment, this DVD polisher device has been reconfigured. Particularly, the distance between the center axes of both the upper DC electric motor and the lower rotatable plate is increased. Moreover, the height of the upper fabric circular plate, with respect to the lower rotatable plate, is also increased. Then, the configured device in Figure 5 can be used to polish the surface of the MC 901 spur gear. To do that, the MC 901 gear is placed on top of the lower rotatable circular plate. The circular fabric polishing pad, which is driven by the upper DC electric motor, is adjusted to contact directly on the desired treating surface. The distance between the pad and the gear side is adjustable by a screw to control the operation speed of the rotational base. To enhance the polishing quality, the four polishing liquids denoted in Figure 5 were also applied directly on the gear-treating surface.

The polishing procedure contains the initial raw treatment step, fine treatment step, and polishing step. Initially, a type of 800 grit (called *No.* 800) sanding paper in a circular shape is attached to the DC electric motor spindle. By combining this sanding paper and the polishing liquid No. 1 in Figure 5, the condition of the MC 901 gear side surface changed dramatically from the original roughness value of *R_z_* ≈ 142 µm to the new roughness value of *R_z_* ≈ 60.2 µm. In this initial step, the operational speed of the bottom rotational base, when measured by a laser measurement tachometer, was approximately 1170 rpm, and the operational time was in 30 s. Next, the sanding paper is replaced by the fabric polishing pad for the fine treatment step. In this research, the fine treatment step had stopped after every 30 s for the measurement of the roughness values via the Olympus LEXT OLS5000—the digital three-dimensional measuring laser microscope. The fine treatment processing is done after six times of stops. In this step, the polishing liquid *No.* 2 was applied to the MC 901 gear surface for the first four times of stops while, for the rest remaining two times, the polishing liquid *No.* 3 was used. At the end of this second treatment process, the roughness value *R_z_* ≈ 60.2 µm had changed considerably to a new value of *R_z_* ≈ 10.8 µm. Then, the polishing step was conducted to finish the gear surface treatment procedure. In this final process, the polishing liquid No. 4 is used. The measurement of the surface quality proceeded after every 30 s at the beginning and 120 s in the end. However, the measured data illustrate that there is not much change occurring in the roughness value, since the final measurement indicates that the roughness value of *R_z_* is almost equal to 10 µm. This 10 µm value is mostly equal to the roughness value measured at the end of the second step of the process—the fine treatment step. In other words, the roughness value at the end of the fine treatment step remains mostly constant until the end of the surface treatment of the MC 901 gear.

Figure 6 specifies the roughness chart of the accomplished polishing process. In this figure, the blue line denotes the roughness value *R_z_* while the orange line denotes the arithmetical mean roughness *R_a_* value. At a first glance, the sharp change of the roughness value *R_z_* right after the initial treatment has occurred between the “Raw” and “Sanded” indexes can be observed. The fine treatment step belongs to the range between “Sanded” and 210 s. Although a reduction in the roughness value is noticeable, the change becomes small in value compared to the change of the surface condition in the initial treatment step. Moreover, it is certain that the roughness value *R_z_* is almost stable when the treatment process surpasses 210 s and reaches 720 s or about 8.5 min in total at around 10 µm. On the contrary, the arithmetical mean roughness *R_a_* does not show any critical change in its value during the treatment procedure.

Three practical surface conditions of the MC 901 gear during the surface treatment process are specified in Figure 7. In this figure, the raw surface of the MC 901 gear in Figure 7a has numerous circular grooves on its surface. These grooves probably are the result of the gear manufacturing technology. As discussed earlier, this type of surface is unsuitable not only for direct spraying of the NPS-J conductive ink, but also for applying the polyimide tape layer. Figure 7b shows a remarkable change in the surface quality after the sanding process with a No. 800 grit sandpaper. The final treated surface is presented in Figure 7c, where its surface quality is qualified sufficiently via empirical work for the direct spraying of the conductive ink and the laser sintering process. The *R_z_* value is around 10 µm.

#### 2.2.2. Experimental Setting and Procedure for the Static Phase Fluctuation

The experimental setting in Figure 8 is for the consideration of the coupling distance effect and the static phase fluctuation effect. Since both the coupling distance effect and the static phase fluctuation effect are conducted by only one experimental rig, the experimental setting must be configured compatibly to use for two cases without any additional configuration during the test. To do that, a magnetic base with an arm and a three-dimensional (3D) stage is employed to combine with each other. While the purpose of the magnetic base is used to hold the monitoring antenna firmly and in parallel with the smart gear, the 3D stage will carry and perform two movements of the smart gear. For the first case, the experiment of the coupling distance effect requires the movement of the smart gear along its central axis or the Z direction. Then, for the second case, the experiment of phase fluctuation requires the rotation of the smart gear around this Z direction at a fixed distance; here, it is 1 mm. Prior to the start of the experiments, the monitoring antenna—held by the arm of the magnetic base and the smart gear—is carried by the three-dimensional stage and must be aligned concentrically and in parallel with each other. This can be considered to be the most important and time-consuming initial step, since it can directly affect the experimental results if any faults occur during the configuration. According to [11], to retain the magnetic coupling of the system, the monitoring antenna must be kept connected to the network analyzer from the beginning until the end of the experiment via its integrated RF connector. In addition, to prevent or mitigate any unexpected noise that might occurr due to the loose connecting point during the experiment, the connecting point of the monitoring antenna and the network analyzer should be secured. Here, the connector point is consolidated by a non-metallic tape, as illustrated in Figure 8a. Figure 8b,c shows the 3D stage, which has a top plate that can be moved to the X, Y, and Z direction and rotated around its center axis. As a result, the two movements of the smart gears in the experiments including the rotation for the phase fluctuation and the translation along the center axis for the coupling distance analyses can be conducted via this 3D stage.

Figure 8a,b provides a complete experimental configuration of the monitoring antenna and the smart gear for the coupling distance and static phase fluctuation effect experiments. The monitoring antenna is fixed concentrically and in parallel with the smart gear while it is connected to the network analyzer. As discussed in the previous paragraph, the alignment of the monitoring antenna and the smart gear before the experiments is crucial to the reliability of the final experimental results. Therefore, the tips to avoid or minimize the potential faults that can occur during the experimental setting is the configuration for the smart gear, and the 3D stage should be performed first. The 3D stage is required to take place on a flat surface, and the smart gear must be fixed firmly to the rotatable top plate of this 3D stage. Then, the monitoring antenna is placed directly on the surface of the smart gear, the sintered sensor surface, for the concentric alignment. In this way, the distance between the monitoring antenna and the smart gear is equal to 0 mm. Therefore, it can be stated that the monitoring antenna is in parallel with the smart gear surface. Next, the concentrical alignment for the central points of the monitoring antenna pattern and the smart gear sensor pattern has proceeded at 0 mm of distance. Once the central points of the monitoring antenna and the smart gear are centered, this relative position of the monitoring antenna, with respect to the smart gear, can be locked immediately thanks to the flexibility of the articulating arm of the magnetic base. Now, the experiment setting is ready for the experiment. In addition, Figure 8a also shows that both the monitoring antenna and the smart gear are attached indirectly to the articulating arm of the magnetic base and to the 3D stage via two dielectric-plastic tubes. The bonding between these tubes and the monitoring antenna or smart gear is also made by durable double tape. In this way, the magnetic coupling of the monitoring antenna and smart gear can occur without any effect from the nearby metallic components.

In the first step of the coupling distance effect experiment, the experimenter locks the rotatable top plate and the X and Y direction of translational movement, so that only the Z direction is available for the experiment. The rotatable top plate is locked at 0 degrees of the relative phase angle between the patterns of the smart gear and the monitoring antenna. Next, from the position of a 0 mm distance between the aligned monitoring antenna and the smart gear, the top plate of the 3D stage is gradually decreased downward. By doing this, the coupling distance value can be increased. In this research, the translational movement in the Z direction of the smart gear ranges from 0 mm to 6 mm. The incremental distance is selected as 0.05 mm for the first range of 0 mm to 2 mm and as 0.25 mm for the second range of 2 mm to 6 mm. These incremental values are controlled by a high-precision knob with a resolution of 0.01 mm of the 3D stage denoted in Figure 8c. Then, during the increase in the coupling distance, the resonant return loss of the system is captured via the network analyzer for each specific distance value. The frequency range of the network analyzer is limited to the range of 0 GHz to 1.0 GHz instead of 0 GHz to 8.0 GHz. The reason for this is referred to in [11], which indicated that the range of 0 GHz to 1 GHz contains the first valley on the resonant return loss chart of the magnetic coupling system, valued at around 0.3 GHz. Practically, there are two valleys that appear on the resonant return loss chart at around 0.3 GHz and 0.9 GHz in the frequency range from 0 GHz to 1 GHz; the 0.3 GHz is found in the most significant and valuable resonant frequency value for the gear failure detection process. Moreover, the setting of the frequency range from 0 GHz to 1.0 GHz of the network analyzer can alleviate the capacity of the measured experimental data compared to the range from 0 GHz to 8.0 GHz. Prior to the experiment, the network analyzer must be calibrated to assure the integrity of the experimental data. Since the return loss data of an antenna is a scatter data type, 1000 points of data were selected for the experiments in this section as a required setting of the network analyzer. To save the experimental data of this experimental work to a computer, it is necessary to use software, namely Anritsu Shock Line, which was provided by the network analyzer manufacturer.

Turning to the phase fluctuation effect experiment, this experimental setting requires the Z translational direction to be locked at 1 mm and the rotatable top plate to be relieved. In reference to Figure 8b, the rotatable top plate has 360 dividers that are equivalent to 360 degrees. Hence, the original phase angle between the monitoring antenna and the smart gear is configured to match the divider of 0 degrees. To perform the experiment on the static phase fluctuation effect, the phase angle values are changed regularly. In this research, the incremental value is 5 degrees. Simultaneously, the resonant return loss data of the monitoring antenna are captured by the network analyzer. The selected frequency range of the network analyzer is also from 0 to 1.0 GHz. Once the phase angle turns to the 360th divider, the experiment is accomplished.

#### 2.2.3. Experimental Setting and Procedure for High-Speed Phase Fluctuation

Figure 9 illustrates a test rig modified for evaluating the phase fluctuation effect at high speed. Overall, the main part of this device is a high-speed motor that can speed up from 500 rpm to 3500 rpm. To confirm the accuracy of the speed value, a tachometer-model DT2230 of Lutron is used for the confirmation. The spindle of the motor is placed horizontally since the coupling model of the monitoring antenna and the smart gear is set horizontally. To fix the smart gear to the spindle head, a plastic nut was made and adjusted with high accuracy to allow the smart gear to be operated stably in parallel with the monitoring antenna at its high speed. In contrast to the smart gear, the monitoring is held on a holder that is translatable along the smart gear’s central axis thanks to an equipped rail at the bottom. Since the holder and the rail are two original parts of the device, they are all metallic parts. Hence, similar to the role of the plastic nut that is made to keep a distance between the metallic spindle shaft and the smart gear, a plastic tube has been used to install between the holder and the monitoring antenna. This technique aims to alleviate the effect of metal on the magnetic coupling and the resonant return loss of the system. Moreover, this type of configuration enables the coupling distance to be changed simply with a minimum error that might occur to the parallelism of the two magnetic coupled components.

In this research, a similar monitoring antenna and smart gear are used for both the static phase fluctuation effect and the high-speed phase fluctuation effect tests. Nevertheless, unlike the experiments described in Section 2.2.2, where the coupling distance and the phase fluctuation are considered separately, these two separate experiments are combined in the high-speed phase fluctuation experiment. Specifically, at each specific coupling distance in the range of 1 mm to 6 mm, the effect of high-speed phase fluctuation is evaluated at the full range of rotational speed from 0 rpm to 3500 rpm. By combining the phase fluctuation and the coupling distance experiments in this section, the author’s purpose is to clarify the relevance between these two factors—the coupling distance and the speed of the smart gear and the resonant return loss of the system. In this way, it is simple to conclude which are the best parameters of the coupling distance and the gear operation speed for the smart gear system to operate more properly, even under the effect of the phase fluctuation. Theoretically, the effect of phase fluctuation on a magnetic coupling system is unavoidable or comprehensively removable. Similar to the previous sections, the resonant return loss data of the phase fluctuation effect at a high-speed test of the smart gear health monitoring system is also collected by the network analyzer. Here, the authors have used a right-angle RF low-loss cable for the connection of the monitoring antenna to the network analyzer. To avoid noises to be added up to the targeted signal, this connection is also secured and kept stable thanks to the non-metallic durable tape. Moreover, according to the experimental setting in Section 2.2.2, the selected frequency range of this device is also limited from 0 to 1.0 GHz, with 1000 points of scatter signal values. The advantage of this selection of 1000 points is that it can help reduce the interval time of the device to capture the data, since the device only has to scan a short range of frequency.

Prior to the operation of the phase fluctuation effect at the high speed of the smart gear of this section, the experimental devices are required to be calibrated. Initially, the coupling distance is set at 1.0 mm. The value of the coupling distance is measured and controlled by a Mitutoyo digital caliper model CD67-S20PS. At 0 rpm of the speed of the smart gear, the resonant return loss of the monitoring antenna is measured and saved as the original data for the coupling distance of 1.0 mm. These original return loss data are considered valuable if they are measured when the phase angle between the smart gear pattern and the monitoring antenna pattern is aligned to 0 degrees. Fundamentally, based on this original resonant return loss data, the authors can evaluate the percentage of change occurring in the resonant return loss data of the magnetic coupling system due to the various values of the speed of the smart gear. Next, the motor of the horizontal spindle is turned on to reach its initial speed of 500 rpm. During the rotation of the smart gear at this 500 rpm speed, the value of return loss is also measured and saved for the 500 rpm speed at 1.0 mm. Similarly, the remaining resonant return loss values of the smart gear system can be measured and saved after each 500 rpm of the speed value. Once the speed of the smart gear reaches its maximum speed at 3500 rpm, the final resonant return loss signal is measured and the test rig will be stopped. That is, the procedure for obtaining the resonant return loss values of the smart gear system operating at a speed range of 0 rpm to 3500 rpm with the coupling distance of 1.0 mm. By repeating this procedure for the remaining coupling distance from 2 mm to 6 mm, the experiment of the phase fluctuation effect at high speed can be finished. Results are shown in the next section.

## 3. Results

In this section, the relevant experimental results are shown. The results include the effect of coupling distance and phase fluctuation resonant return loss data of the smart gear system at the static condition and the phase fluctuation resonant return loss data at the high speed of the smart gear.

### 3.1. Coupling Distance

Figure 10 shows the return loss data for the effect of the coupling distance experiment. The data consists of single monitoring antenna return loss and coupled return loss between the monitoring antenna and the smart gear. The return loss of the single monitoring antenna is the green solid line that has a one-peak shape valley in the vicinity of the 0.3 GHz one. This one-peak shape valley changes to a two-peak shape valley when the magnetic coupling occurs. At a first glance, the results of the resonant return loss in this figure agree with the discussion in the previous Section 2.2.2 about the 0.3 GHz valley of the resonant return loss shape. Corresponding to the change of the coupling distance value, the valleys at 0.3 GHz of the resonant return loss values have evidently shown changes in their shapes compared to the 0.9 GHz valleys’ shapes. Overall, the one-peak shapes of the 0.9 GHz valleys mostly remain unchanged while the shapes of the 0.3 GHz valleys visually shift between the two-peak shapes and the one-peak shapes when the coupling distance values have been increased from 0 mm to 6 mm. Consequently, the valleys of the resonant return loss at 0.3 GHz are focused only for the analysis in Figure 11.

According to Figure 11, it is clear that the changing amplitude of the resonant return loss values is much different for three minor ranges of distance. These are the ranges of 0 mm to 1 mm, 1 mm to 2 mm, and 2 mm to 6 mm. In these three ranges, the maximum changing amplitude of the resonant return loss values can be seen in the first range of 0 mm to 1 mm, especially from 0 mm to 0.5 mm. Specifically, the depth of the second resonant peak located on the right side of the 0.3 GHz valley in Figure 11a shows a remarkable changing amplitude of −1.82 dB. However, within the range of 0 mm to 0.5 mm, the changing amplitude also reaches −1.222 dB, which takes about 67.14 [%] of the total changing amplitude value in the 0 mm to 1 mm range. In Figure 11b, within the range of 1 mm to 2 mm, the changing amplitude is insignificant for the second resonant peak on the right side, while this value remains constant for the first resonant peak. The resonant peaks at 1 mm and 2 mm are denoted by the blue line and red line, respectively. The noticeable point in this figure is the variation of the valleys’ shapes from the two-peak shape at 1 mm to the approximate one-peak shape at 2 mm of the coupling distance. Figure 11c clearly shows the change of the valleys’ shapes into a one-peak shape at 6 mm. This one-peak shape is denoted by the red line. The changing amplitude of the peaks at 2 mm and 6 mm is equal to 0.694 dB.

### 3.2. Static Phase Fluctuation

Figure 12 shows the resonant return data of the static phase fluctuation experiment. As discussed in Section 2.2.2, the coupling distance on which this experiment has been conducted is 1 mm. Based on the result in Figure 12, apparently, the change of the return loss signal occurs at both valleys—in the vicinity of 0.3 GHz and 0.9 GHz. However, the 0.9 GHz valley was ignored, since its one-peak shape valley is less important than the two-peak shape valley of the resonant return loss value at 0.3 GHz. This 0.3 GHz two-peak shape valley is focused for analysis in Figure 13.

As can be seen, Figure 13 indicates that the forms of the resonant return loss data have been clearly changed in accordance with the switching of the phase angle values. However, these changes do not occur proportionally to the increase in the phase angle values from 0 degrees to 360 degrees. Specifically, in the range of 0 degrees to 180 degrees of the phase angle values, the return loss values of the first resonant peak have steadily changed from their minimum value of −9.03 dB to a maximum value of −8.80 dB. Moreover, the frequency value of this peak also decreases from a maximum value of 0.296 GHz to a minimum value of 0.294 GHz. In the range of 0 degrees to 360 degrees, the return loss values and the frequency values of the second resonant peak also change from their maximum value of −7.21 dB to a minimum value of −7.42 dB and their maximum value of 0.343 GHz to a minimum value of 0.342 GHz. Overall, in the first half of the 360 degrees of the phase angle values, the frequency values of two resonant peaks gradually decrease, while the values of the return loss of these peaks increase and decrease, respectively. Likewise, in the second half of the 360 degrees of the phase angle values, the changes also appear for both the return loss and frequency values. The ranges of return loss and frequency for the first resonant peaks are −9.04 dB; 0.295 GHz to −8.82 dB; 0.294 GHz and the second resonant peaks are −7.20 dB; 0.343 GHz and −7.41 dB; 0.342 GHz. Surprisingly, in contrast to the tendency of the change in the return loss and the frequency value of the first half of 360 degrees of the phase angles, the return loss value decreases for the first peak and increases for the second peak, while the frequency values of these two peaks gradually increase.

### 3.3. High-Speed Phase Fluctuation

Figure 14 provides the experimental results of the high-speed phase fluctuation experiment. In this figure, there is a total of 54 values of the resonant return loss data collected during the smart gear operations at high speed. Generally, there is not much difference between the typical shapes of the resonant return loss data measured at high speed and at static conditions. The most distinguished point is that the increase in the speed of the smart gear results in the numerous minor peaks appearing on the resonant return loss data of the magnetic coupling system. Moreover, it is simple to recognize that the density of minor peaks nearby the 0.3 GHz valley, is much higher than in the other places. In addition, the 0.3 GHz valley also has noticeable shifts in the resonant return loss signal compared to the 0.9 GHz valley. Hence, this 0.3 GHz valley was used for the analysis and evaluation.

In Figure 15 and Figure 16, the smoothness of the resonant return loss data charts is noticed. At the first glance, this smoothness of the resonant return loss apparently changes correspondingly to the change of the coupling distance. From Figure 15, the resonant return loss charts of the system measured at 6 mm become the smoothest charts compared to other charts in the range from 1 mm to 6 mm of the coupling distance. Basically, the typical shapes of the resonant return loss data that are two-peak and one-peak shape valleys are still recognizable, respectively. By observing the different groups of resonant return loss data charts measured at different coupling distances, the tendency of the 0.3 GHz valley shapes of the resonant return loss data shows that the change from two-peak to one-peak shape valleys depends on the coupling distance instead of the rotational speed of the smart gear. In addition, Figure 15 and Figure 16 consider the relevance of these resonant return loss data, the coupling distance, and the rotational speed of the smart gear in detail based on the smoothness of the resonant return loss data.

Figure 16 contains eight minor figures of the extracted return loss data at a specific speed and in a range of a coupling distance. The arrangement of the component figures in Figure 16 is based on the increase in the rotational speed. Specifically, from Figure 16a–h, the collection of the resonant return loss values that have a similar rotational speed and range from 0 rpm to 3500 rpm with an increment of 500 rpm are presented, respectively. Moreover, at each of these specific rotational speeds and one minor figure, six resonant return loss values that belong to six values of coupling distances are provided for the comparison, respectively. These six values are denoted by six different colors. The blue color line presents the return loss value measured at the coupling distance of 1 mm. Similarly, the other resonant return loss values measured at the coupling distance of 2, 3, 4, 5, and 6 mm are correspondingly presented by magenta, green, brown, dark green, and red color lines.

As can be seen, Figure 16a exemplifies six different original resonant return loss datasets that are all measured when the speed of the smart gear is equal to 0 rpm and the phase angle between patterns of the monitoring antenna and the smart gear is aligned to 0 degrees. At a first glance, the resonant return loss charts of six resonant return loss values show that there are no minor peaks appearing on the charts of these signals. Due to the increase in the coupling distance from 1 mm to 6 mm, the two-peak shape valley at 0.3 GHz is gradually transformed into the one-peak shape valley at 0.3 GHz, of the resonant frequency of the resonant return loss data. In this case, the obtained results agree with the results of the previously considered experiment of the coupling distance effect denoted in Figure 10 and Figure 11 in this section. Moreover, compared to Figure 15, the arrangement of resonant return loss results based on the coupling distance in each minor figure in Figure 16 enables the effect of phase fluctuation at various speeds on the resonant return loss at each specific coupling distance value to be observed clearly. For example, the observation of only the resonant return loss measured data at 1 mm of the coupling distance—the blue line from Figure 16a–h—calculates the undeniable effect of the speed of the smart gear on the resonant return loss signal of the system. In other words, in accordance with the increase in the speed value, the state of the smooth resonant return loss chart measured at the speed of 0 rpm has steadily changed to the unsmooth resonant return loss chart measured at the speed of 3500 rpm. The higher the rotational speed is, the more unsmooth the resonant return loss chart is, and vice versa. Similarly, the effects of phase fluctuation at various speeds of the smart gear on the resonant return loss data measured at 2 mm to 6 mm of the coupling distance can be recognized simply. Furthermore, these findings also prove that, although the speed of the smart gear has an effect on the resonant return loss data of the magnetic coupling system, this effect also depends on the coupling distance. Specifically, by looking at one minor figure, for example, Figure 16h, where the resonant return loss data are measured at a similar speed of 3500 rpm of the smart gear but at different values of the coupling distance, the smoothness of the resonant return loss charts changes from an unsmooth state of the blue line to the smoother state of the red line. This means that the magnitude of the effect is reduced when the coupling distance is increased and vice versa.

## 4. Discussion

In reference to the results in the Section 3, the results of the static coupling distance test in Figure 10 and Figure 11 are useful for the authors to be able to select the most suitable coupling distance value between the monitoring antenna and the smart gear for the effectiveness of a smart gear health-monitoring system. In the scope of this research, the health condition of the smart gear is healthy in normal working conditions. Moreover, the coupling distance for the phase fluctuation experiment at static speed conditions and high-speed conditions is empirically selected as 1 mm. As referred to in [IBA SPIE], the results showed that the healthy condition of the smart gear results in a great magnetic coupling effect. In other words, the condition of the magnetic coupling between the smart gear and the monitoring antenna in this research should show a clear two-peak shape valley at 0.3 GHz of the resonant frequency of the resonant return loss chart. In reference to the experimental results of the coupling distance provided in Figure 11, the values of the coupling distance in the range of 0.25 mm to 0.30 mm are ideal for producing a two-peak shape valley of the resonant return loss signal. Nevertheless, due to the current limitation of our developing smart gear system, this ideal range of coupling distance cannot be afforded. The limitation comes from the substrate material of the monitoring antenna. As introduced in Section 2, the substrate material of the monitoring antenna is POM plastic. Due to the properties of plastics, thermal expansion is one of the most concerning issues. Specifically, since the friction between a master gear and the operation gear (the smart gear during the gear meshing process is unavoidable for an oilless gear meshing system), it is believed that the temperature of the gear can increase remarkably. Hence, the coupling distance of 0.25 mm to 0.3 mm is currently too risky to be applied for a dynamic smart gear meshing system, such as in this research. In addition, the experimental coupling distance results in Figure 11 also demonstrate that the amplitude of variation of the return loss value in the coupling distance range of 0 mm to 0.75 mm, especially in the range of 0 mm to 0.5 mm, is much higher than in the range of 0.75 mm to 1 mm. This discussion means that, if the coupling distance is selected within the 0 mm to 0.75 mm range, or even within the 0 mm to 0.5 mm range, the coupling distance of a practical system should be measured and controlled carefully. Since the error of even 0.05 mm can change the resonant return loss values remarkably, the current smart gear system that depends on POM plastics for the fabrication of the monitoring antenna seems not to be able to control the 0.05 mm error of the distance coupling. The 0.05 mm error can simply come from the current issues of our developing smart gear systems, such as the thermal expansion or the bending of the POM plate of the antenna, due to the additional layer of polyimide. The 0.75 mm to 1.0 mm range can be considered to be the most suitable range of values for the selection of the coupling distance; however, compared to 0.75 mm, the selection of 1 mm as the coupling distance is more favorable for our smart gear system. Indeed, the experimental results of the coupling distance in Figure 11, especially Figure 11a,b, show that the 1 mm coupling distance produces a more stable resonant return loss signal than the 0.75 mm. At 1 mm, the fluctuation of the resonant return loss value is insignificant, and even the tolerance of the coupling distance can be ±0.05 mm. Moreover, in a practical driving test, the 1 mm coupling distance also shows its advantages in the prevention of stuck objects between the smart gear and the monitoring antenna. Here, the stuck objects are frequently known as the debris of broken gears at the end of the driving test.

Turning to the static phase fluctuation experiment, the results in Figure 12 and Figure 13 evidently indicate that the resonant return loss signals are significantly changed in line with the change in the phase angle values. In fact, the resonant return loss values measured at 0 degrees and measured after 360 degrees are similar. In other words, after 360 degrees from the starting point at 0 degrees, the phase angle value returns to its initial position, and the resonant return loss measured at the starting point also returns to its original value. Hence, this finding affirms that the change of the phase angles triggers the shift of the resonant return loss signals and the frequency values of the resonant peaks. Moreover, in reference to our previous research by [12], where a phase fluctuations test was conducted for two POM plate antennas, the result of this research specified that the change of the phase angles between two antennas results in the fluctuation of the resonant return loss signal. The second POM antenna in this previous research was shorted by a short module, therefore its characteristic condition is equivalent to the characteristic condition of the smart gear in this research, with a healthy and normal conductive sensor. Nevertheless, in contrast to the results provided in Figure 12 and Figure 13, the change of the frequency values at resonant peaks in the case of two POM plates is insignificant or mostly constant. The reason for this, the difference is believed to come from the fact that the sintered antenna patterns of two POM plate antennas are placed symmetrically with each other, while the patterns of the monitoring antenna and smart gear in this current research are placed asymmetrically. This kind of arrangement makes the direction of the two spiral coils opposite to each other in the case of two POM plates antennas and similar to each other in the case of the monitoring antenna and smart gear in this research. In this way, the characteristics and direction of the magnetic field generated by the network analyzer antenna side are apparently different for the two cases. Moreover, the appearance of a sensor chain only on the smart gear pattern in this research is also believed to be a factor contributing to the magnetic coupling data at each certain phase angle. Overall, the fluctuations of the measured return loss values in Figure 12 and Figure 13 are unavoidable for a magnetic coupling system; however, it is also no longer an issue in our developing smart gear system. Specifically, according to Figure 12, the maximum fluctuating value of the return loss is about 0.23 dB for the first half and 0.22 dB for the second half of 360 degrees. This difference even does not change the typical two-peak form of the magnetically coupled signal, therefore the method provided in [13] is still useful for identifying the necessary parameters of the smart gear wirelessly.

Regarding the high-speed phase fluctuation test results in Figure 14, Figure 15 and Figure 16, the influence of the phase fluctuation at high speed relates majorly to the smoothness of the resonant return loss chart. In fact, the results prove that, when the speed of the smart gear increases, there are minor peaks appearing on the resonant return loss of the system. The density of the minor peaks is seen to be high around the peaks of the resonant return loss signal chart. Although the experimental results in this research also prove that the high operational speed is not a significant concern to our current developing smart gear system, the results are also meaningful to some extent. As described in Section 2, the developing smart gear system works based on the wireless magnetic coupling of a monitoring antenna and a smart gear—the working gear. Moreover, the resonant return loss data collected by the monitoring antenna is pivotal to the gear failure detection analysis. However, since the density of minor peaks on the resonant return loss signal increases with the increase in the rotational speed, the risk of making a wrong diagnosis about the physical condition of the gear is potential. Therefore, a necessary solution should be proposed to improve the gear health-monitoring method. Moreover, Figure 15 and Figure 16 obviously indicate that, in comparison with the results of the static phase fluctuation test in Figure 12, the typical two-peak form of the magnetic coupling resonant return loss of a monitoring antenna and a healthy smart gear is still able to be observed. Since the two-peak shape valley of the resonant return loss benefits the wireless gear failure detection, it is still able to detect the physical condition of the smart gear. Moreover, the results of the phase fluctuation at high speed are useful for the authors to evaluate the empirical selection of 1 mm as the coupling distance. In reference to the experimental results of the coupling distance in Figure 12 and the relevance between the coupling distance and speed of the smart gear on the resonant return loss of the system in Figure 16, the results in Figure 16 indicate that the effect of a high-speed gear on the resonant return loss of a magnetic coupling system can be reduced when the coupling distance is increased. This means that the obtained results in Figure 16 could be affected more severely due to the high speed of the smart gear if the selected coupling distance is smaller than 1 mm. This finding, on one hand, consolidates our empirical selection of 1 mm for the coupling. On the other hand, it indicates the potential issues that can occur to the resonant return loss of the system when the values of coupling distance become smaller and smaller.

## 5. Conclusions

This paper has considered the effect of the phase angle between a monitoring antenna and a smart sensor gear on the resonant return loss data in our developing smart gear system. Here, the phase angle is the relative angle between two open spiral coils of the monitoring antenna and smart gear patterns. The experimental components include a monitoring antenna and a healthy smart gear. The effect of the phase angle focuses on two states of the smart gear: the static condition and a high speed. In the first case, the obtained experimental results of the static phase fluctuation experiment indicate that the change of the relative phase angle affects the resonant return loss of the magnetic coupling system—our smart gear system. Moreover, compared to the static phase fluctuation experiment, the high-speed phase fluctuation experiment that enables the speed of the smart gear to be changed from 0 rpm to 3500 rpm also proves that the switching speed of the phase angle relates directly to the smoothness of the resonant return loss signal of the smart gear system. Moreover, the high-speed phase fluctuation experiment also considers both the effects of distance and speed of the smart gear simultaneously. The results of this experiment indicate that the magnitude of the phase fluctuation effect on the resonant return loss signal depends on the value of the coupling distance. In other words, the effect can become more severe or less severe when the coupling distance value is reduced or increased.

The coupling distance of 1 mm is applied for the static phase fluctuation experiment. This distance is known as the distance between the monitoring antenna and the smart gear. Since this 1 mm coupling distance has been selected based on the empirical experience of the authors, a coupling distance experiment has been conducted to evaluate this pre-selected coupling distance. Via the experimental results and discussion, the coupling distance of 1 mm has proven that this value is suitable for the application of the current developing smart gear system. Moreover, this 1 mm coupling distance is also consolidated as a result of the high-speed phase fluctuation. Specifically, at 1 mm of the coupling distance, although the received results of the high-speed phase fluctuation experiment have been affected by the high speed of the phase angle switching, it still remains the typical two-peak shape valley on the resonant return loss signal. This return loss, therefore, still benefits the gear failure detection process.

In fact, the considered problems in this research are essential to the improvement of our system. Since the resonant return loss is the only unique data of the developing smart gear system and the effect of the fluctuation of phase angle is also unavoidable, knowing the degree of this effect on the resonant return loss could encourage the authors to propose compatible solutions to prevent or reduce the undesirable effect if necessary. As an example, the return loss data that is received from the high-speed phase fluctuation experiment poses a challenge to the authors. This challenge is not only for the parameter identification process, but also for the gear failure detection. Once a corresponding method is found to deal with this challenge, it is believed that the accuracy will be improved significantly. The smart gear used in this research involved a healthy gear instead of an unhealthy gear. Hence, in the next step, the authors plan to consider the experiment for both a healthy (normal gear) and an unhealthy fault gear so that a more comprehensive evaluation could be proposed for our developing smart gear system.

## Figures and Tables

**Figure 1 sensors-22-03231-f001:**
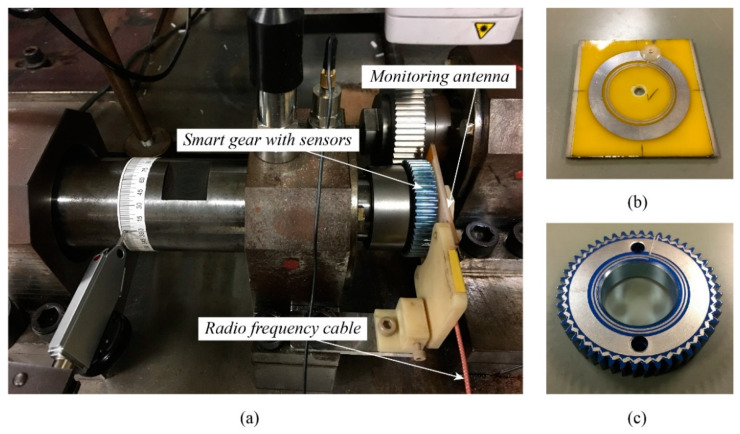
The smart gear health monitoring system: (**a**) practical smart gear health monitoring system; (**b**) monitoring antenna; (**c**) smart gear sensor.

**Figure 2 sensors-22-03231-f002:**
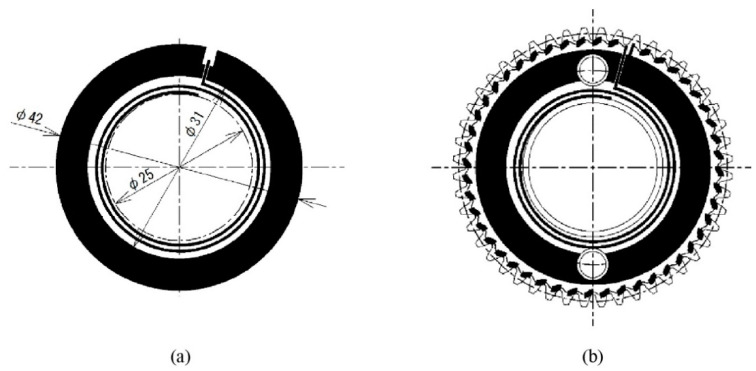
The laser sintering patterns: (**a**) monitoring antenna; (**b**) sensor antenna.

**Figure 3 sensors-22-03231-f003:**
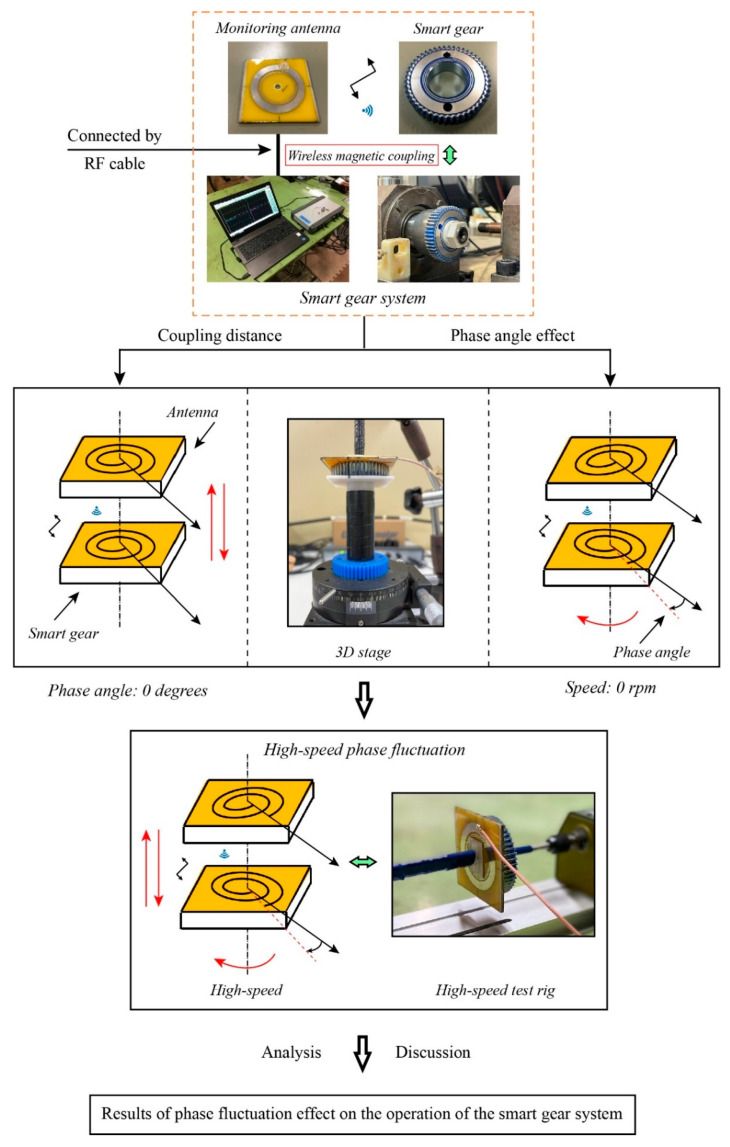
The procedure of the experiments. This figure shows the flowchart of the research.

**Figure 4 sensors-22-03231-f004:**
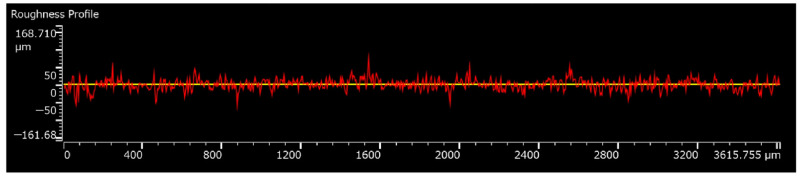
The roughness profile of a raw MC 901 gear.

**Figure 5 sensors-22-03231-f005:**
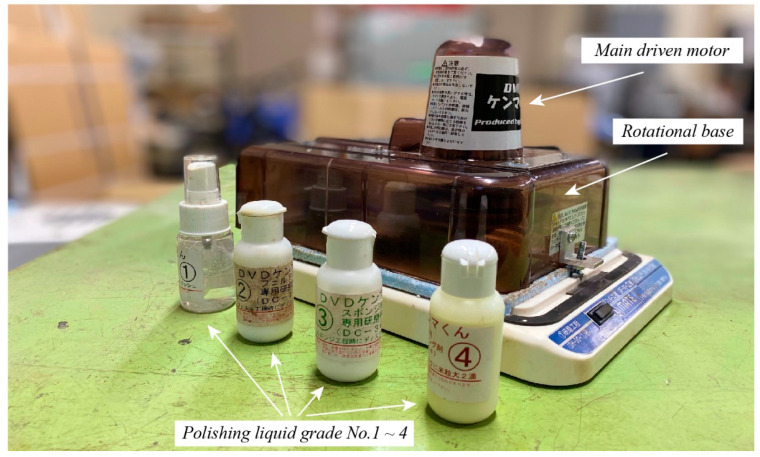
A self-configured plastic gear polisher. Originally, this is a DVD polisher by Kenmac.

**Figure 6 sensors-22-03231-f006:**
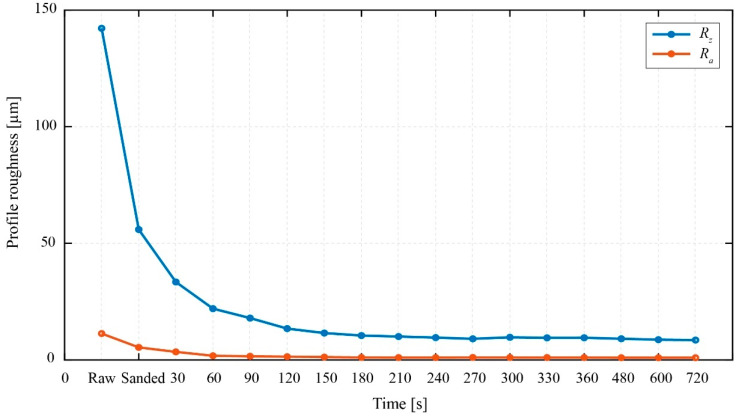
The roughness value *R_z_* of the MC 901 gear during the treatment process.

**Figure 7 sensors-22-03231-f007:**
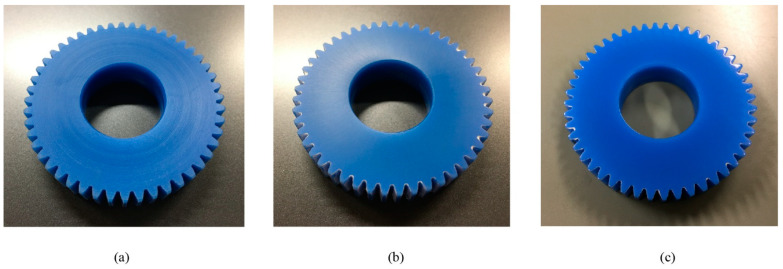
The comparison of the surface quality of an MC 901 gear before and after the treatment procedure process: (**a**) raw surface (original gear surface); (**b**) sanded surface; (**c**) finished surface.

**Figure 8 sensors-22-03231-f008:**
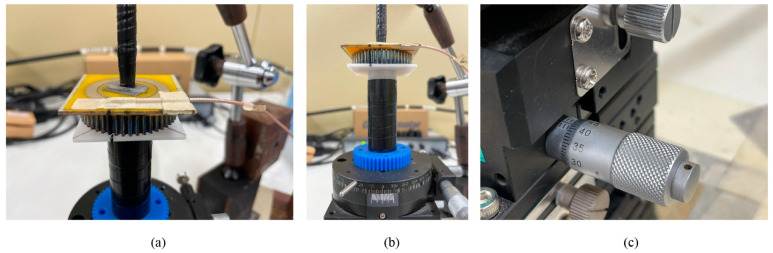
The experimental setting for the phase fluctuation effect at the static condition: (**a**) general setting; (**b**) top rotational plate is locked at 0 degrees of phase angle; (**c**) high precision knob with a ratchet.

**Figure 9 sensors-22-03231-f009:**
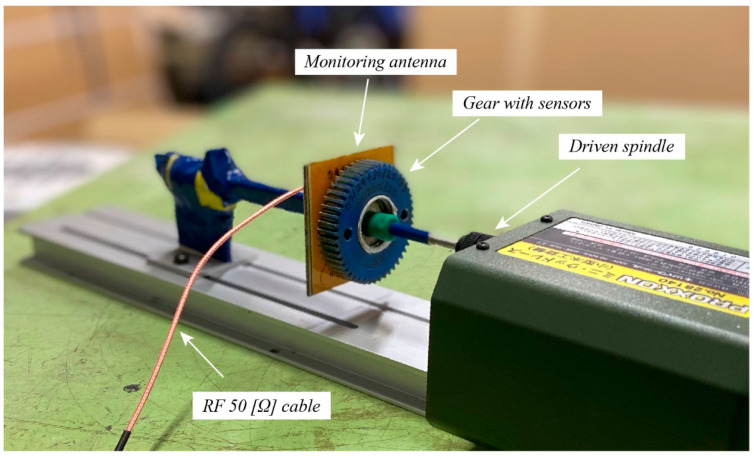
The self-configured device and setting for the experiment of phase fluctuation effect at high speed. Originally, this is a mini lathing machine by Proxxon.

**Figure 10 sensors-22-03231-f010:**
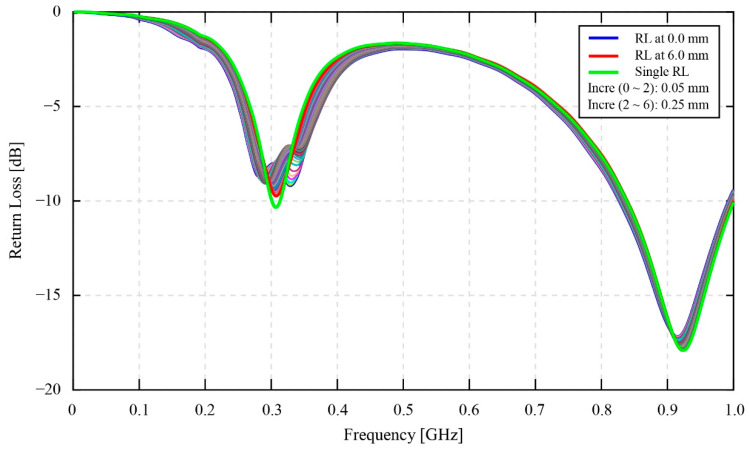
The effect of the coupling distance on the resonant signals of the smart gear system.

**Figure 11 sensors-22-03231-f011:**
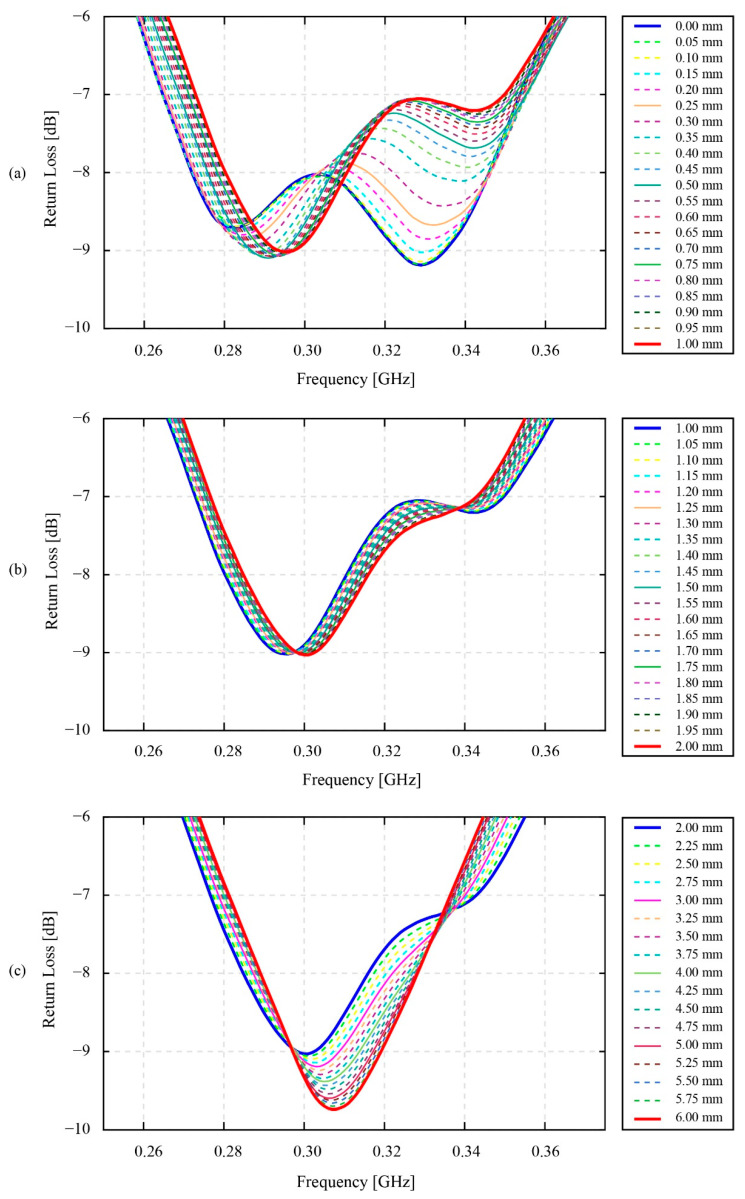
The effect of coupling distance on the resonant signals of the smart gear system: (**a**) from 0 to 1 mm, increment of 0.05 mm; (**b**) from 1 to 2 mm, increment of 0.05 mm; (**c**) from 2 to 6 mm, increment of 0.25 mm.

**Figure 12 sensors-22-03231-f012:**
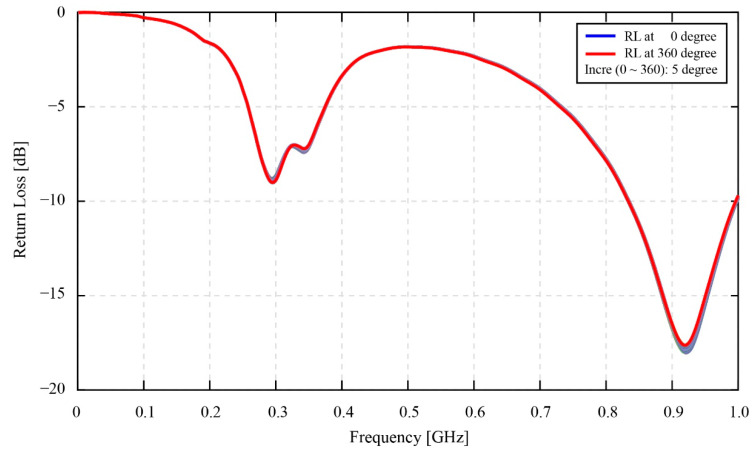
The effect of phase fluctuation on the resonant signals of the smart gear system in static conditions.

**Figure 13 sensors-22-03231-f013:**
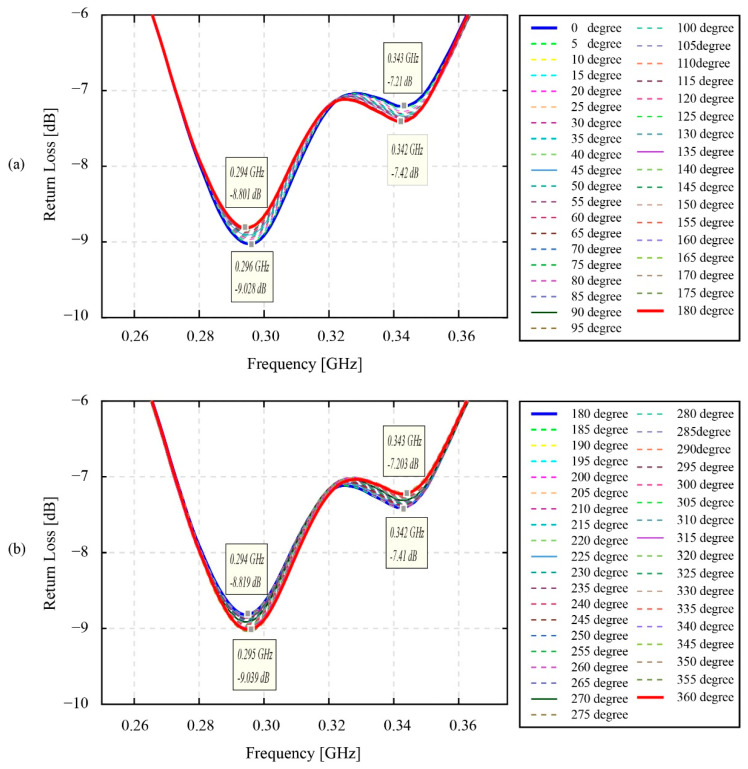
The effect of phase fluctuation on the resonant signals in the static conditions in the range of 0.2–0.4 GHz: (**a**) range of 0 degrees to 180 degrees; (**b**) range of 180 degrees to 360 degrees.

**Figure 14 sensors-22-03231-f014:**
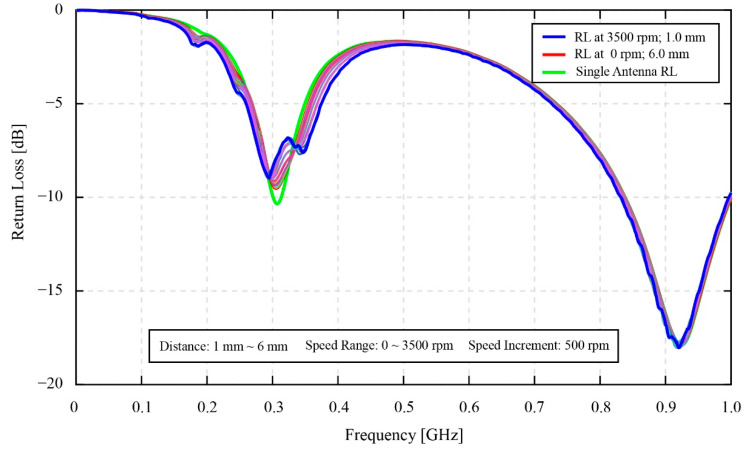
The effect of phase fluctuation at the high speed of the smart gear.

**Figure 15 sensors-22-03231-f015:**
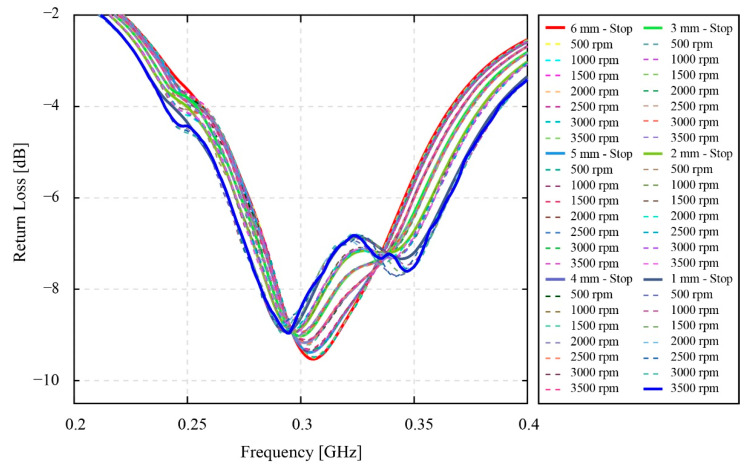
The effect of phase fluctuation at the high speed of the smart gear (limited to the vicinity of 0.3 GHz valley).

**Figure 16 sensors-22-03231-f016:**
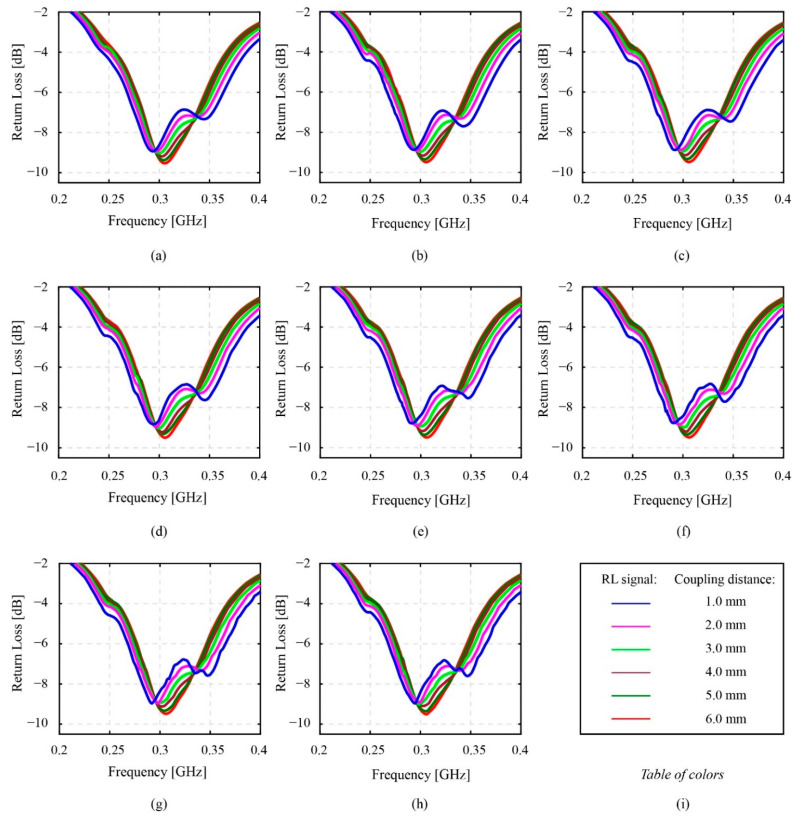
The effect of phase fluctuation at various speed from Stop - 0 rpm to 3500 rpm with coupling distance from 1 mm to 6 mm (limited to the vicinity of 0.3 GHz valley): (**a**) Stop−0 rpm, 1−6 mm; (**b**) 500 rpm, 1−6 mm; (**c**) 1000 rpm, 1−6 mm; (**d**) 1500 rpm, 1−6 mm; (**e**) 2000 rpm, 1−6 mm; (**f**) 2500 rpm, 1−6 mm; (**g**) 3000 rpm, 1−6 mm; (**h**) 3500 rpm, 1−6 mm; (**i**) description of chart lines.

## Data Availability

The data presented in this study are available on request from the corresponding author.

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
