# Peer review of "Effect of Phase Fluctuation on the Proper Operation of Smart Gear Health Monitoring System"

_sensors, 2022, doi:10.3390/s22093231_

Round 1

Reviewer 1 Report

In this work, the authors are trying to present the effect of coupling distance and phase fluctuation at the static condition, and high speed of smart gear on the return loss of receiver antenna, which is important in industry applications. However, there exists several issues that is necessary to be solved before it is accepted. Below are the listed suggestions and revise comments.

  1. The current title is not quite clear on the motivation and main significance of this work. Instead, the authors were showing the specific main contents, which may not help on understanding from readers.

  1. The authors has mentioned in the manuscript that the “effect” of coupling distance and phase fluctuation. However, there is no clear conclusion found in the related part of the manuscript: what is the specific effect of them?

  1. The materials and mechanism of the sensing from figure 1 to figure 4 should be presented: What is the material of antenna and its base material? What is the mechanism of sensing, electromagnetic effect? As is known, the distance effect in electromagnetic effect is clear in theory. 

  1. The presentation of “smart” “healthy” are not clear: what is the definition of smart and healthy of structure? Additionally, the conclusion of 1 mm is not rigor: there is not discussion on the theoretical support. If this result is merely from experimental results, then a tricky problem comes: why the author considers the 1 mm, instead of 1.05 mm? Additionally, why the data 0.99 mm, or 1.01 is not experimentally considered? A strict and logical analysis should be given on this issue. Besides, there seems to be no discussion or conclusion on the phase.

Author Response

RESPONSE TO THE REVIEWER #1

First of all, we would like to express our gratitude for the reviewer’s meaningful comments, which helped us significantly improve our paper.

Comment 1: The current title is not quite clear on the motivation and main significance of this work. Instead, the authors were showing the specific main contents, which may not help on understanding from readers.

Response: Thank you very much for your recommendation. We agree with your indicated issue. At present, we have considered and changed the current title to: “Effect of phase fluctuation on the proper operation of smart gear health monitoring system.”

In this title, the terms “static condition” and “high speed of smart gear” are removed. Moreover, the term “coupling distance” is also deleted.

The reasons that support this change are: 

  • The motivation of the research is to evaluate the influence of the phase fluctuation between the smart gear and the monitoring antenna in our developing smart gear health monitoring system. This system works based on the magnetic coupling which relates directly to the resonant return loss of the monitoring antenna. This resonant return loss data is important for the operation of our smart gear system. Due to the theory of magnetic field, the change of the relative phase angles of the two spiral coils – one belongs to the monitoring antenna and the other belongs to the antenna of the smart gear is believed to have effects on the magnetic coupling signal between these two coils. Hence, it can result in the change of the resonant return loss of the monitoring antenna. It means that the phase fluctuation can be a factor that can affect the proper operation of the smart gear health monitoring system to some extent.
  • The “static condition” and “high speed of smart gear” all belong to the concept of phase fluctuation. The “static” and “high speed” present only two states of the “phase fluctuation”. So, they can be replaced by just “phase fluctuation.”
  • The “coupling distance” has been removed compared to the previous title according to our discussion. In fact, due to the configuration of the experiment for the high-speed test of the smart gear, the distance of 1 [mm] is selected based on the practical experimental device to ensure the smooth and safe operation of the smart gear at high speed. This distance also shows that it can adapt very well to our driving test rig as in Fig. 1. Specifically, now, the substrate material of the monitoring antenna is polyacetal plastic (POM). And during a practical driving test, the temperature of the smart gear certainly increases due to the gear meshing between that gear and a master gear. On one hand, the temperature of the POM plate monitoring antenna is potentially increased drastically if the distance between this antenna and the smart gear is too close. As a result, the substrate POM material of the monitoring antenna can be suffered remarkably from thermal expansion, especially since this POM plate has a hole at its center for a nut to fix the smart gear to the spindle head. Because of this design, a severe thermal expansion can make the POM plate monitoring antenna become unparallel with the smart gear surface or even contact with the smart gear surface and damage the sensor layer of the smart gear. Therefore, the consideration of the coupling distance in the manuscript majorly aims to indicate how the return loss can be influenced due to the distance. And it also shows us how the quality of the coupling return loss can be changed in the range of 0 to 6 [mm] where 1 [mm] belongs.

Fig. 1 A smart gear system in a driving test rig.

Comment 2: The authors has mentioned in the manuscript that the “effect” of coupling distance and phase fluctuation. However, there is no clear conclusion found in the related part of the manuscript: what is the specific effect of them?

Response: Thank you very much for your comment. We beg your understanding about the missing conclusion of the effect of the coupling distance. The effect of coupling distance has been proposed in the manuscript. However, according to the “response for comment 1”, the purpose of the experimental result focuses on figuring out the variation of the resonant return loss signal corresponding to the change of the coupling distance. Our experimental work also shows that the return loss value and its shape have been changed when the coupling distance is changed. In other words, it is also able to conclude empirically that the coupling distance has an impact on the resonant return loss of the magnetically coupled system. And based on the result of the variation of the resonant return loss signal, we can observe and evaluate the resonant return loss value of the distance in the vicinity of 1 [mm] – the pre-selected distance for the experimental configuration and device.

Comment 3: The materials and mechanism of the sensing from figure 1 to figure 4 should be presented: What is the material of antenna and its base material? What is the mechanism of sensing, electromagnetic effect? As is known, the distance effect in electromagnetic effect is clear in theory. 

Response: Thank you very much for your suggestions and questions. We have added more explanation and information to clarify your concern.

What is the material of antenna and its base material? 

As described in the “response for comment 1”, the base material of the antenna is polyacetal plastic or POM. Since POM plastic material is also the material of plastic gear in our laboratory, we have selected POM as the based material for the antenna. In this way, we can mitigate the unexpected effect due to different materials when we conduct the pairing between the POM monitoring antenna and a POM smart gear. Now, the material of the smart gear has been changed from POM to MC 901 nylon material. The reason is that the mechanical properties of the MC 901 nylon are much higher than the POM gear. However, the material of the monitoring is kept intact – the POM plastic. Currently, we are considering sintering the conductive ink on a metal plate to make an antenna on this material.

The material of the antenna layer is similar to the material of the sensor on the smart gear. To make an antenna, a laser sintering technique is used to sinter the silver nanoparticle ink on the surface of the POM plate. This surface is covered by one layer of polyimide tape to improve the surface condition for the higher quality of the sintered layer.

Fig. 2 Smart gear system: (a) Monitoring antenna (b) Smart gear

 What is the mechanism of sensing, electromagnetic effect?

Figure 2 shows two components of our developing smart gear system. Basically, the designs of the antenna patterns on the monitoring antenna and on the smart gear are identical. The difference between these two sintered patterns of the monitoring antenna and the smart gear is the sensor chain that surrounds the antenna pattern on the smart gear.

As can be seen, the sensor change is designed to cover all the gear tooth root area of the smart gear – here are 48 [teeth]. The two ends of the sensor chain of the smart gear pattern are connected to the open spiral coil and the annular band called the ground (GND) part of an antenna, respectively. By this design, since the sintered open spiral coil, and GND part are conductive, these two parts and the sensor chain form a close sensor track circuit. The conductivity of this close sensor track circuit depends mostly on the conductivity of the sensor chain. Turning to the monitoring antenna, this antenna pattern also has the spiral coil and the GND part. These two parts are connected to two poles of the attached radio frequency (RF) connector on the monitoring antenna, Fig. 2(a). And when this RF connector is connected to a network analyzer to obtain the return loss of the monitoring antenna, the network analyzer, the spiral coil, and the GND part will also form an antenna circuit. Our experimental works in [1] have found that, when the smart gear with its close sensor track circuit is placed nearby the monitoring antenna that also has the antenna circuit, the magnetic coupling occurs. The result of the magnetic coupling is the change in the return loss of the antenna circuit. Specifically, the return loss of the described antenna circuit that has a one-peak valley at around 0.3 [GHz] changes to be a resonant return loss that has a two-peak valley at around 0.3 [GHz].

During the gear operation, if cracks appear at any root area of any tooth, the conductivity of the sensor chain will be affected – mostly will be reduced. As a result, the conductivity of the close sensor track circuit will be reduced. Hence, the magnetic coupling between the antenna circuit and the close sensor track circuit will be affected. This effect certainly can make a change to the resonant return loss of the monitoring antenna. Practically, the return loss shape is changed. Therefore, based on the observation of this resonant return loss shape during the gear operation, the physical condition of the sensor chain or the smart gear base material can be monitored. Results of [1] indicated two forms of the resonant return loss had been obtained in correspondence with two physical states of the sensor chain: cracked – unhealthy and uncracked – healthy. This defines the mechanism of the sensing technique via the magnetic electronic coupling effect.

We also agree with you about “the distance effect in electromagnetic effect is clear in theory”. In our previous study [2], the magnetic diagram in Fig. (3) has been analyzed. In this diagram, R1 , L1 , C1 are the resistance, inductance, and capacitance of the monitoring antenna circuit. R2, L2 , C2 are the respective parameters of the close sensor track circuit. es, Z0  are the power supplied by the network analyzer and characteristic impedance of this device. I1 , I2  are the currents flowing in the two closed-loop circuits, the monitoring antenna circuit, and the close sensor track circuit. The mutual inductance occurred at the space where the two resonant circuits couple with each other is denoted as L12.

Fig. 3: Magnetic coupling diagram of the smart gear system

In fact, the mutual inductance L12 of the coupling system in Fig. 3 will be influenced significantly due to the distance effect. Particularly, the distance is related to the magnitude of the magnetic coupling. This can be observed clearly via the result of the coupling distance presented in the manuscript. The result indicates that when the distance is increased, the return loss shape tends to change from a two-peak valley into a one-peak valley and vice versa. The one-peak valley is closed to return loss shape on the single antenna – without coupling. So, it means that the magnitude of magnetic coupling reduces remarkably when the coupling distance increases and vice versa. Besides, the distance between the monitoring antenna and the smart gear in our smart gear system is often fixed constantly during the gear operation. Therefore, the distance effect in our developing smart gear system depends merely on the designed coupling distance, here is 1[mm]. The experimental results have proved that at this 1 [mm], the resonant return loss shape remains its two-peak valley shape. This shape is important for the wireless condition monitoring of the smart gear in our gear health monitoring technique.

Comment 4: The presentation of “smart” “healthy” are not clear: what is the definition of smart and healthy of structure? Additionally, the conclusion of 1 mm is not rigor: there is not discussion on the theoretical support. If this result is merely from experimental results, then a tricky problem comes: why the author considers the 1 mm, instead of 1.05 mm? Additionally, why the data 0.99 mm, or 1.01 is not experimentally considered? A strict and logical analysis should be given on this issue. Besides, there seems to be no discussion or conclusion on the phase.

Response: Thank you very much for your questions. We beg your understanding of the unclear explanation.

We would like to respond to your inquiries as follows:

The presentation of “smart” “healthy” are not clear: what is the definition of smart and healthy of structure? 

The term “smart” implies that the operation gear can become “smart” when we integrate the laser sintering sensor on its surface. Specifically, as described above, any crack or break that occurs on the operation gear integrated with the sensor will result in the transform of the resonant return loss shape of the monitoring antenna. Hence, compared to a normal gear meshing system, the role of the sensor can make our gear meshing system become smarter. That is for the definition of “smart”.

Regarding the term “healthy”, this term aims to describe the “uncracked” state of the operation gear integrated by the sensor. This comes from the technical term “structure health monitoring”. So, the “healthy” structure is equivalent to the normal structure – without any problems or errors.

Additionally, the conclusion of 1 mm is not rigor: there is not discussion on the theoretical support. If this result is merely from experimental results, then a tricky problem comes: why the author considers the 1 mm, instead of 1.05 mm? 

Thank you for your interesting question. We beg your understanding about the conclusion of 1 [mm]. According to the “response for comment 1” above, the selection of 1 [mm] is mostly based on our experimental work and empirical experience. The experimental result of coupling distance helps us to be able to consolidate the selection of 1[mm]. Based on our experimental work, the distance of 1 [mm] shows its advantages such as being simple to adjust and fix the distance or it is wide enough to keep the temperature of the antenna remarkably lower than the temperature of the smart gear. Although the distance of 0.5 [mm] or 0.75 [mm] indicates that the two-peak shape of the resonant signal is clearer than at 1 [mm], this distance requires shorter periods of time to supervise the system to control the temperature and accidence due to the overheat. In the future when this system is ready to be applied to the industry, the coupling distance might be able to be reduced to some extent thanks to the high technology of manufacturing or new material. However, at present, 1 [mm] is one of the most reliable values of coupling distance for the test.

The reasons for not considering the distance of 1.05 mm are partially based on our experimental initial setting. Since the measurement of the return loss at the values around 1 [mm] did not provide any noticeable points, the 1 [mm] coupling distance seems to be the most favorable value for the experimental configuration. Moreover, the experimental results in this research also indicate that compared to the range of 0 [mm] to 0.5 [mm] and 0.5 [mm] to 0.75 [mm], the change of the return loss value is pretty hard to figure out between the coupling distance values of 1 [mm] and 1.05 [mm]. Hence, the selection of 1[mm] becomes more convenient for the experimental configuration.

Additionally, why the data 0.99 mm, or 1.01 is not experimentally considered? A strict and logical analysis should be given on this issue.

Honestly, the resonant return loss data includes the range of 0 to 1 [mm] with a step of 0.01 [mm] has been conducted before in an experiment. The result of this experiment is illustrated in Fig. 4. The initial purpose of consideration of the coupling distance with an increment of 0.01 [mm] is to clarify the variation of resonant return loss data in detail in the range of 0 [mm] to 1[mm], especially the range from 0[mm] to 0.5 [mm].

Besides, as can be seen in Fig. 5(a), the 0 [mm] to 0.5 [mm] range induces remarkable changes in the shape of the resonant return loss. Although the experimental sample of the experiment with results in Fig. 4 and the experiment with results in Fig. 5 are different, the massive variation of resonant return loss value of the system also occurs in the range of 0 [mm] to 0.5 [mm]. And out of this range, especially from 0.75 [mm] and above, the variation of the resonant return loss has been reduced noticeably. In scrutiny of Fig. 4 and Fig. 5 (a), it gradually becomes hard to observe the change of the resonant return loss value of the system when the coupling distance increases from 0 [mm] to 1[mm]. Especially in the vicinity of 1 [mm], the change is almost not available. Figure 5(a) also proves that even the increment is 0.05 [mm], the resonant return loss value measured at 0.95 [mm] and 1.00 [mm] is approximately identical. Moreover, compared to Fig. 5(a), Fig. 5(b) also reveals that the amplitude of changes in values on the resonant return loss of Fig. 5(a) is much higher than Fig. 5(b) even though both distances are equal to 1[mm] with 0.05 [mm] of increment.

Based on this analysis, within the range of 0 [mm] to 1 [mm] or even to 2 [mm], the data of resonant return loss at 0.99 [mm] or 1.01 [mm] can be eliminated and replaced by the return loss at 1 [mm] since the amplitude of changes is unrecognizable via figures and chart. Besides, the purpose of considering the coupling effect in this research is to indicate how the resonant return loss responds to the change in the coupling distance. Hence, the requirement of obtaining the return loss data with the increment of 0.01 [mm] might be unnecessary. This also explains the selection of 0.05 [mm] of increment in the coupling distance experiment that provides results in Fig. 5.

Fig. 4: Coupling distance from 0 [mm] to 1[mm] with increment of 0.01 [mm]

Besides, there seems to be no discussion or conclusion on the phase.

Thank you very much for your comment. We will check and add for information for the discussion part and conclusion part on the phase.

We hope that this answer may find you well. If you have any more questions, please feel free to leave your comments. We will try to give the answers as well as possible!

Fig. 5: Coupling distance from 0 [mm] to 6[mm] with increments of 0.05 [mm] and 0.25 [mm]

REFERENCES

  1. Iba, D.; Futagawa, S.; Miura, N.; Iizuka, T.; Masuda, A.; Sone, A.; and Moriwaki, I. Development of smart gear system by conductive-ink print (Impedance variation of a gear sensor with loads and data transmission from an antenna). Proc. SPIE 10973, Smart Structures and NDE for Energy Systems and Industry 4.0 1097309, Denver, Colorado, USA (04-05 Mar 2019).
  2. Mac, T. T.; Iba, D.; Matsushita, Y.; Miura, N.; Iizuka, T.; Masuda, A.; Sone, A.; and Moriwaki, I. Identification of Q value of smart gear antenna by return loss analysis of receiving antenna (Non-contact health monitoring system for gears manufactured by printing conductive ink). The Proceedings of the Dynamics & Design Conference2020:436, Niigata, Japan (01-04 Sept 2020).

Reviewer 2 Report

  1. The title looks like the thesis title must be short as the research title
  2. Please change the word study to research as it is represented like a thesis.
  3. Please provide some statistical results at the end of the abstract in the conclusion
  4. The first paragraph of the introduction is very long and has references. Please make sure to provide references and make it short as well.
  5. The second paragraph of the introduction is very long with the same references, it would be better to make it short.
  6. The last part of the introduction looks like taken from some thesis, please rewrite and convert it to a research paper The content of this manuscript is based on the following structure: 
    Chapter 1 – Introduction: Motivation and objective of this study. 
    Chapter 2 – Materials and methods: Explanation of materials and methods used in this paper. 
    Chapter 3 – Results and discussion: Experimental results, analysis, and discussion. 
    Chapter 4 – Conclusion
  7. The method is understandable and explained very well
  8. The repetition of the reference in most of the paper-like references [14]
  9. The results are explained too long,  please make it a short explanation of each figure. Also, the figures are not in the same sequence some wrote Figure and some are Fig. please make it according to the numbers as well.
  10. The conclusion is very long and there are no statistical results in the conclusion as well.

Author Response

RESPONSE TO THE REVIEWER #2 

First of all, we would like to express our gratitude for the reviewer’s meaningful comments, which helped us significantly improve our paper.

Comment 1: The title looks like the thesis title must be short as the research title.

Response: Thank you very much for your recommendation. We agree with your opinion. At present, we have considered and changed the current title to: “Effect of phase fluctuation on the proper operation of smart gear health monitoring system.”

In this title, the terms “static condition” and “high speed of smart gear” are removed. Moreover, the term “coupling distance” is also deleted.

The reasons that support this change are: 

  • The motivation of the research is to evaluate the influence of the phase fluctuation between the smart gear and the monitoring antenna in our developing smart gear health monitoring system. This system works based on the magnetic coupling which relates directly to the resonant return loss of the monitoring antenna. This resonant return loss data is important for the operation of our smart gear system. Due to the theory of magnetic field, the change of the relative phase angles of the two spiral coils – one belongs to the monitoring antenna and the other belongs to the antenna of the smart gear is believed to have effects on the magnetic coupling signal between these two coils. Hence, it can result in the change of the resonant return loss of the monitoring antenna. It means that the phase fluctuation can be a factor that can affect the proper operation of the smart gear health monitoring system to some extent.
  • The “static condition” and “high speed of smart gear” all belong to the concept of phase fluctuation. The “static” and “high speed” present only two states of the “phase fluctuation”. So, they can be replaced by just “phase fluctuation.”
  • The “coupling distance” has been removed compared to the previous title according to our discussion. In fact, due to the configuration of the experiment for the high-speed test of the smart gear, the distance of 1 [mm] is selected based on the practical experimental device to ensure the smooth and safe operation of the smart gear at high speed. This distance also shows that it can adapt very well to our driving test rig as in Fig. 1. Specifically, now, the substrate material of the monitoring antenna is polyacetal plastic (POM). And during a practical driving test, the temperature of the smart gear certainly increases due to the gear meshing between that gear and a master gear. On one hand, the temperature of the POM plate monitoring antenna is potentially increased drastically if the distance between this antenna and the smart gear is too close. As a result, the substrate POM material of the monitoring antenna can be suffered remarkably from thermal expansion, especially since this POM plate has a hole at its center for a nut to fix the smart gear to the spindle head. Because of this design, a severe thermal expansion can make the POM plate monitoring antenna become unparallel with the smart gear surface or even contact with the smart gear surface and damage the sensor layer of the smart gear. Therefore, the consideration of the coupling distance in the manuscript majorly aims to indicate how the return loss can be influenced due to the distance. And it also shows us how the quality of the coupling return loss can be changed in the range of 0 to 6 [mm] where 1 [mm] belongs.

Fig. 1 A smart gear system in a driving test rig.

Comment 2: Please change the word study to research as it is represented like a thesis.

Response: Thank you very much for your suggestion. We have changed it in the manuscript.

Comment 3: Please provide some statistical results at the end of the abstract in the conclusion.

Response: Thank you very much for your suggestion. Regarding this question, the last part of the abstract has stated that the findings of this research are meaningful to the authors for evaluating and improving the accuracy of this gear health monitoring technique. In fact, the resonant return loss signal is the essential data in our developing smart gear system. Hence, knowing the change of the return loss due to the phase fluctuation, specifically when it is compared with the resonant return loss measured without any effect of the phase fluctuation (at a specific static phase angle of the magnetic coupling), the percentage of this difference can suggest the author several solutions. For example, to select the best parameters such as the speed value or the coupling distance for the gear to minimize the error of the return loss value.

We have added the results and several discussions to the conclusion of the manuscript.

Comment 4: The first paragraph of the introduction is very long and has references. Please make sure to provide references and make it short as well.

Response: Thank you very much for your comment. We have checked and re-written it to make it as short as possible in the manuscript.

Comment 5: The second paragraph of the introduction is very long with the same references, it would be better to make it short.

Response: Thank you very much for your suggestion. We have modified it in the manuscript.

Comment 6: The last part of the introduction looks like taken from some thesis, please rewrite and convert it to a research paper The content of this manuscript is based on the following structure:

- Chapter 1 – Introduction: Motivation and objective of this study.

- Chapter 2 – Materials and methods: Explanation of materials and methods used in this paper.

- Chapter 3 – Results and discussion: Experimental results, analysis, and discussion.

- Chapter 4 – Conclusion

Response: Thank you very much for your suggestion. We have rewritten and converted it to become suitable for a research paper.

Comment 7: The method is understandable and explained very well.

Response: Thank you very much for your comment and compliment. We will try our best. 

Comment 8: The repetition of the reference in most of the paper-like references [14]

Response: Thank you very much for your comment. We have modified this in the manuscript.

Comment 9: The results are explained too long,  please make it a short explanation of each figure. Also, the figures are not in the same sequence some wrote Figure and some are Fig. please make it according to the numbers as well.

Response: Thank you very much for your comment and suggestion. We have checked and made the explanation of the results shorter. Incidentally, the problem with the “Figure” and “Fig.” has been fixed. 

Comment 10: The conclusion is very long and there are no statistical results in the conclusion as well.

Response: Thank you very much for your comment. We have modified it to make the conclusion more concise. The statistical results have also been added and discussed.

We hope that this answer may find you well. If you have any more questions, please feel free to leave your comments. We will try to give the answers as well as possible!

Reviewer 3 Report

The article presents a smart sensor system for condition monitoring of gear. Interesting experimental research is presented. However, it is not known how the developed system can be used to monitor working systems without disassembling the gear.

Author Response

RESPONSE TO THE REVIEWER #3

First of all, we would like to express our gratitude for the reviewer’s meaningful comments, which helped us significantly improve our paper.

Comment 1: The article presents a smart sensor system for condition monitoring of gear. Interesting experimental research is presented. However, it is not known how the developed system can be used to monitor working systems without disassembling the gear.

Response: Thank you very much for your comment and question. In order to answer your question, we would like to add Figure 1 below.

Fig. 1 Smart gear system: (a) Monitoring antenna (b) Smart gear

Figure 1 shows two components of our developing smart gear system. Basically, the designs of the antenna patterns on the monitoring antenna and on the smart gear are identical. The difference between these two sintered patterns of the monitoring antenna and the smart gear is the sensor chain that surrounds the antenna pattern on the smart gear.

As can be seen, the sensor change is designed to cover all the gear tooth root area of the smart gear – here are 48 [teeth]. The two ends of the sensor chain of the smart gear pattern are connected to the open spiral coil and the annular band called the ground (GND) part of an antenna, respectively. By this design, since the sintered open spiral coil and GND part are conductive, these two parts and the sensor chain form a close sensor track circuit. The conductivity of this close sensor track circuit depends mostly on the conductivity of the sensor chain. Turning to the monitoring antenna, this antenna pattern also has the spiral coil and the GND part. These two parts are connected to two poles of the attached radio frequency (RF) connector on the monitoring antenna, Fig. 1(a). And when this RF connector is connected to a network analyzer to obtain the return loss of the monitoring antenna, the network analyzer, the spiral coil, and the GND part will also form an antenna circuit. Our previous experimental works [1] have found that, when the smart gear with its close sensor track circuit is placed nearby the monitoring antenna that also has the antenna circuit, the magnetic coupling occurs. The result of the magnetic coupling is the change in the return loss of the antenna circuit. Specifically, the return loss of the described antenna circuit that has a one-peak valley at around 0.3 [GHz] changes to be a resonant return loss that has a two-peak valley at around 0.3 [GHz].

During the gear operation, if cracks appear at any root area of any tooth, the conductivity of the sensor chain will be affected – mostly will be reduced. As a result, the conductivity of the close sensor track circuit will be reduced. Hence, the magnetic coupling between the antenna circuit and the close sensor track circuit will be affected. This effect certainly can make a change to the resonant return loss of the monitoring antenna. Practically, the return loss shape is changed. Therefore, based on the observation of this resonant return loss shape during the gear operation, the physical condition of the sensor chain or the smart gear base material can be monitored. Results of [1] indicated two forms of the resonant return loss had been obtained in correspondence with two physical states of the sensor chain: cracked – unhealthy and uncracked – healthy. This defines the mechanism of the sensing technique via the magnetic electronic coupling effect.

Figure 2 introduces a practical model of the smart gear health monitoring system in a driving test rig. As can be seen, the operation gear – the blue gear engages with a master gear for the power transmission. This operation gear is called a smart gear since it has a sintered layer of the sensor on its surface. The monitoring antenna is designed to be able to be fixed in front of the surface where the sensor layer is located of this gear. This design aims to make sure that the magnetic coupling between the monitoring antenna and the smart gear can be kept even if the smart gear is working or stopped. Therefore, the characteristic condition of the operation gear – a smart gear can be monitored continuously without disassembling the gear.

Fig. 2 A practical smart gear health monitoring system

REFERENCES 

  1. Iba, D.; Futagawa, S.; Miura, N.; Iizuka, T.; Masuda, A.; Sone, A.; and Moriwaki, I. Development of smart gear system by conductive-ink print (Impedance variation of a gear sensor with loads and data transmission from an antenna). Proc. SPIE 10973, Smart Structures and NDE for Energy Systems and Industry 4.0 1097309, Denver, Colorado, USA (04-05 Mar 2019).

We hope that this answer may find you well. If you have any more questions, please feel free to leave your comments. We will try to give the answers as well as possible!

Round 2

Reviewer 1 Report

The mentioned questions and suggestions are solved and answered to meet the standard of acceptence. The current version of the manuscript is suggested to be accepted for publication.